# Genome-Wide Identification of the ABC Gene Family and Its Expression in Response to the Wood Degradation of Poplar in *Trametes gibbosa*

**DOI:** 10.3390/jof10020096

**Published:** 2024-01-24

**Authors:** Jia Zhao, Achuan Wang, Qian Wang

**Affiliations:** 1College of Computer and Control Engineering, Northeast Forestry University, Harbin 150040, China; zjwork2021@163.com; 2Department of Computer Science, Durham University, Durham DH1 3LE, UK; qian.wang173@hotmail.com

**Keywords:** ABC gene family, wood degradation, *Trametes gibbosa*, genome-wide analysis, RT-qPCR analysis

## Abstract

Wood-rotting fungi’s degradation of wood not only facilitates the eco-friendly treatment of organic materials, decreasing environmental pollution, but also supplies crucial components for producing biomass energy, thereby reducing dependence on fossil fuels. The ABC gene family, widely distributed in wood-rotting fungi, plays a crucial role in the metabolism of lignin, cellulose, and hemicellulose. *Trametes gibbosa*, as a representative species of wood-rotting fungi, exhibits robust capabilities in wood degradation. To investigate the function of the ABC gene family in wood degradation by *T. gibbosa*, we conducted a genome-wide analysis of *T. gibbosa*’s ABC gene family. We identified a total of 12 *Tg-ABCs* classified into four subfamilies (ABCA, ABCB, ABCC, and ABCG). These subfamilies likely play significant roles in wood degradation. Scaffold localization and collinearity analysis results show that *Tg-ABCs* are dispersed on scaffolds and there is no duplication of gene sequences in the *Tg-ABCs* in the genome sequence of *T. gibbosa*. Phylogenetic and collinearity analyses of *T. gibbosa* along with four other wood-rotting fungi show that *T. gibbosa* shares a closer phylogenetic relationship with its same-genus fungus (*Trametes versicolor*), followed by *Ganoderma leucocontextum*, *Laetiporus sulphureus*, and *Phlebia centrifuga* in descending order of phylogenetic proximity. In addition, we conducted quantitative analyses of *Tg-ABCs* from *T. gibbosa* cultivated in both woody and non-woody environments for 10, 15, 20, 25, 30, and 35 days using an RT-qPCR analysis. The results reveal a significant difference in the expression levels of *Tg-ABCs* between woody and non-woody environments, suggesting an active involvement of the ABC gene family in wood degradation. During the wood degradation period of *T. gibbosa*, spanning from 10 to 35 days, the relative expression levels of most *Tg-ABCs* exhibited a trend of increasing, decreasing, and then increasing again. Additionally, at 20 and 35 days of wood degradation by *T. gibbosa*, the relative expression levels of *Tg-ABCs* peak, suggesting that at these time points, *Tg-ABCs* exert the most significant impact on the degradation of poplar wood by *T. gibbosa*. This study systematically reveals the biological characteristics of the ABC gene family in *T. gibbosa* and their response to woody environments. It establishes the foundation for a more profound comprehension of the wood-degradation mechanism of the ABC gene family and provides strong support for the development of more efficient wood-degradation strategies.

## 1. Introduction

Poplars, known for their rapid growth, wide distribution, and short genome, among other advantages, have found extensive applications in various fields. Their fast growth makes their wood suitable for biomass energy production, and their lightweight, durable, and easy-to-process characteristics contribute to their widespread use in furniture manufacturing, construction, and the paper industry [1]. *Populus simonii* Carr × *Populus nigra* L., a hybrid variety of poplar, inherits favorable traits from both parent species, exhibiting excellent fast-growing properties and features such as drought resistance, cold resistance, and insect resistance. As a result, it has become a key afforestation species in northern regions [2,3,4]. Known for its rapid growth, it typically reaches commercial timber size in a few years. Its wood generally contains a moderate amount of cellulose, making it well-suited for paper and pulp production. Therefore, it has significant potential in areas such as timber production, the paper industry, and biomass energy production [5,6].

Wood degradation is the gradual breakdown of wood into smaller compounds and organic substances in natural surroundings or specific conditions. This process plays a crucial role in the organic material cycle, contributing to the preservation of ecological balance. Furthermore, it serves as a source of raw materials for biomass energy production, offering a means to reduce reliance on fossil fuels [7,8,9,10]. However, wood, consisting of intricate compounds like lignin, cellulose, and hemicellulose, encounters numerous challenges during degradation. Lignin, the most challenging component to degrade in wood, possesses a complex structure and resists degradation by most biological enzymes due to its multiple chemical bonds. Cellulose, which is relatively more susceptible to degradation than lignin, benefits from its linear structure, facilitating enzymes in breaking β-1,4-glucosidic bonds. Hemicellulose presents a degradation difficulty that falls between lignin and cellulose. These components intertwine, forming a robust and less easily destructible structure, thereby heightening the complexity of wood degradation [11,12,13].

Wood-rotting fungi excel at degrading wood, and wood decomposition constitutes their primary way of life. They efficiently break down wood and plant residues, converting organic matter into forms more readily absorbed by other organisms. This process contributes to the natural recycling of organic materials. Furthermore, these fungi can be employed to treat waste from the wood and pulp industry, reducing environmental pollution. They present a potential for low-energy, pollution-free wood degradation and sustainable energy production [14,15]. *Trametes gibbosa*, as one of numerous wood-rotting fungi, possesses significant wood-degrading capabilities. It can release extracellular enzymes such as manganese peroxidases, lignin peroxidase, and laccase, facilitating the degradation of lignin in wood. Additionally, it produces intracellular glucanases, extracellular glucanases, and xylanases, enabling the breakdown of cellulose and hemicellulose in wood. These enzymes assist *T. gibbosa* in breaking down wood into small molecular compounds, achieving wood degradation [16,17,18,19].

The ABC gene family is widely distributed among plants, insects, and microorganisms. Genes in this family encode ABC transporters, which play a pivotal role in the metabolism of lignin, cellulose, and hemicellulose [20,21]. ABC transporters facilitate the movement of the intermediates or degradation products of lignin and cellulose between intracellular and extracellular spaces. They can create channels on the cell membrane, transporting various degradation enzymes involved in breaking down lignin and cellulose from the intracellular space to the extracellular space or other cell structures, thereby enhancing degradation efficiency. Additionally, they can expel potentially toxic substances from the cell’s products into the external environment, maintaining the stability of the intracellular environment [22,23,24]. Researchers have conducted thorough investigations into the relationship between the ABC gene family and lignin and cellulose. For example, de Lima et al. found that the ABC gene (*SvABCG17*) is a potential transporter of lignin monomers [25]. Yu et al. suggested the potential role of the ABC gene (*PgrABCG14*) in promoting plant growth and lignin accumulation [26]. Xie et al. identified ABC family genes in *Phanerochaete chryso-sporium* and suggested that the transporter protein identified in the *P. chrysosporium* secretomes could play a role in cellobiose and/or cellodextrin uptake [27]. Based on these studies, it is speculated that the ABC gene family may have a significant impact on wood degradation. However, at present, no research has explored the specific functions of ABC family genes in wood degradation.

Here, we focused on the degradation process of *P. simonii* Carr *× P. nigra* L. wood by *T. gibbosa*, aiming to investigate the functions and responses of the ABC gene family, conducting analyses on its structure, predicting functions, and identifying characteristics. Additionally, we conducted a quantitative analysis of *T. gibbosa*’s ABC genes at different time points in both woody and non-woody environments using the RT-qPCR analysis, to elucidate the response of ABC gene family in the process of degrading poplar wood of *T. gibbosa*. This study establishes a solid foundation for wood-rotting fungi to contribute to the production of low-energy, environmentally friendly organic matter and sustainable energy.

## 2. Materials and Methods

### 2.1. Sample Preparation

*Trametes gibbosa* (Pers.) Fr. was provided by the Forest Pathology Laboratory of the Forestry Protection subject at the College of Forestry, Northeast Forestry University. The fungus was cultured in 250 mL conical flasks with 70 mL of LNAS medium (Low-Nitrogen Asparagine Succinic Acid) [28] and 5 mL of 15% glucose, with a pH value set at 4.5. The cultures were maintained at 27 °C under static conditions for 10, 15, 20, 25, 30, and 35 days. For the treatment group, 2 g samples of wood chips (1 × 1 × 5 cm) from *Populus simonii* Carr × *Populus nigra* L. were added to the conical flasks, while the control group had no wood chips. Each treatment condition had three biological replicate samples, and each sample was obtained by combining the mycelia from 5 flasks. After the removal of wood chips, the mycelia were collected and stored in liquid nitrogen for subsequent experimental use.

### 2.2. Identification of ABC Genes in T. gibbosa

Using the Pfam database “http://pfam.xfam.org (accessed on 5 September 2023)” [29], we conducted a search for DNA-binding domain sequences in the ABC genes of *T. gibbosa* (*Tg-ABCs*) (Appendix A). Following this, the *Tg-ABCs* were classified into subfamilies using the CDD database “https://www.ncbi.nlm.nih.gov/Structure/bwrpsb/bwrpsb.cgi (accessed on 5 September 2023)” [30]. The DNA binding domain was subsequently visually compared using WebLogo “http://weblogo.berkeley.edu/logo.cgi (accessed on 6 September 2023)”.

### 2.3. Phylogenetic Relationship and Motif Analyses of Tg-ABCs in T. gibbosa

The protein sequences of the *Tg-ABC* gene family were aligned using MEGA 11.0.13 software [31], and a phylogenetic tree was constructed using the neighbor-joining (NJ) method (Appendix A). To further verify the evolutionary relationship of each protein, we constructed a phylogenetic tree of Tg-ABC proteins using the optimal model (LG+G+F) with the maximum likelihood (ML) method (Appendix A). MEME “http://meme-suite.org (accessed on 26 September 2023)” was used to identify conserved motifs in the Tg-ABC protein sequences. TBtools 2.008 [32] was used to visualize the phylogenetic tree and gene structures of *Tg-ABCs*.

### 2.4. Scaffold Localization and Collinearity Analysis of Tg-ABCs in T. gibbosa

The annotation of *Tg-ABCs* was acquired from JGI databases “https://mycocosm.jgi.doe.gov/Tragib1/Tragib1.home.html (accessed on 28 September 2023)”. *Tg-ABCs* were mapped to the genome [33] of *T. gibbosa*, and the duplication events in *Tg-ABCs* were analyzed using MCScanX in TBtools 2.008. Both of these results were visualized using TBtools 2.008.

### 2.5. Phylogenetic Relationship and Collinearity Analyses of ABC Gene Family

Segmental duplication events and collinearity between *Tg-ABCs* and homologous genes from four species of wood-rotting fungi (*T. versicolor*, *G. leucocontextum*, *L. sulphureus* and *P. centrifuga*) were analyzed using Dual Synteny Plotter. The protein sequences of these four species were obtained from the NCBI databases “https://www.ncbi.nlm.nih.gov/protein (accessed on 8 September 2023)”. A phylogenetic tree was constructed using the neighbor-joining (NJ) method in MEGA. The results were visualized in TBtools.

### 2.6. Other Characteristic Analyses in the Tg-ABCs

ProtParam “https://web.expasy.org/protparam (accessed on 16 September 2023)” was used to predict the physical and chemical properties of the Tg-ABC protein (Table 1). To examine the hydrophobicity of the Tg-ABC protein, we employed ExPASy ProtScale “https://web.expasy.org/protscale (accessed on 16 September 2023)” (Appendix A). TMHMM “http://www.cbs.dtu.dk/services/TMHMM (accessed on 16 September 2023)" was employed to forecast transmembrane helices (Appendix A). For the prediction of phosphorylation sites, Netphos “http://www.cbs.dtu.dk/services/NetPhos (accessed on 16 September 2023)” was used (Appendix A). To anticipate the topological heterogeneity model of the Tg-ABC protein, Protter “http://wlab.ethz.ch/protter/start/ (accessed on 16 September 2023)” was employed (Appendix A).

### 2.7. RT-qPCR Analysis of Tg-ABCs in Response to Wood Degradation by T. gibbosa

We separately extracted total RNA from *T. gibbosa* cultured under woody and non-woody environments for 10, 15, 20, 25, 30, and 35 days using the RNAprep Pure Plant Kit (DP441) from Tiangen Biotech (Beijing) Co., Ltd., Beijing, China and selected *Gpd* as the reference gene. Real-time quantitative reverse transcription PCR (RT-qPCR) was then employed to determine the expression levels of *Tg-ABCs* and the reference gene, and the RTq-PCR was performed following the protocol of a TaKaRa one-step RT-PCR kit from Baori Medical Technology (Beijing) Co., Ltd., Beijing, China. The primer sequences are provided in Appendix A. The 2^−ΔΔCt^ [34] method was applied to calculate the relative expression levels of *Tg-ABCs* in woody environments compared to non-woody environments, as well as at different culture times relative to the 10-day culture.

## 3. Results

### 3.1. Identification of ABC Genes in T. gibbosa

We identified 12 members of the ABC gene family (*Tg-ABCs*) from the genome of *T. gibbosa*, all of which contain the conserved ABC transporter binding domain. Using WebLogo “https://weblogo.berkeley.edu/logo.cgi (accessed on 16 September 2023)”, we aligned the sequences of the conserved domains in the ABC transporter domains of the 12 *Tg-ABCs* (Appendix A), and the alignment results for 105 amino acids are depicted in Figure 1a. Each stack in the figure is composed of different amino acid symbols, and the overall height of the stack reflects the degree of sequence conservation at that position. The alignment results demonstrate a high conservation level of the ABC transporter domain sequences among the 12 *Tg-ABCs*. Specific amino acid positions, namely 11, 81, 82, 83, 91, and 104, exhibit complete amino acid identity, indicating excellent consistency in amino acid types at these positions.

The sequence alignment results for the DNA binding domain (Figure 1b) indicate that among the twelve *Tg-ABCs*, nine of them contain two conserved domains: the ABC transporter and the ABC transporter transmembrane region. Among these, seven belong to the ABCC subfamily, and two belong to the ABCB subfamily. However, in the twelve *Tg-ABCs*, three lack the ABC transporter transmembrane region. Of these genes, both *gene5257* and *gene2690* possess four conserved domains, placing them in the ABCG subfamily, while *gene2753* features an ABC-2 family transporter distinct from other *Tg-ABCs*, categorizing it as part of the ABCA subfamily.

### 3.2. Phylogenetic Relationship and Motif Analyses of Tg-ABCs in T. gibbosa

To elucidate the relationships among members of the ABC gene family in *T. gibbosa*, we constructed an NJ-phylogenetic tree using their protein sequences (Appendix A) and identified 20 motifs within the 12 *Tg-ABCs* (Figure 2).

The results indicate that genes classified within the same ABC subfamily cluster on proximate branches, sharing similar motif types. Motif 10, found in all 12 *Tg-ABCs*, is characterized as the ABC transporter-like ATP-binding domain. Genes in the ABCC subfamily exhibit a higher number of motifs ranging from 13 to 19. Genes in the ABCB subfamily have six to seven motifs. The ABCA and ABCG subfamily genes contain the fewest motifs, with each subfamily having only two.

### 3.3. Scaffold Localization and Collinearity Analyses of Tg-ABCs in T. gibbosa

By analyzing the scaffold localization of the 12 *Tg-ABCs*, we discovered that only *gene5243*, *gene5257*, and *gene5291* share the same location on scaffold14, and *gene2690* and *gene2753* are found together on scaffold5, while the rest of the genes are situated on distinct scaffolds individually. This finding indicates a dispersed distribution of *Tg-ABCs* within the scaffolds of *T. gibbosa* (Figure 3a).

We analyzed the segmental duplication events of *Tg-ABCs* in *T. gibbosa* and found no duplication of gene sequences. Additionally, genes located on the eight scaffolds (1, 2, 5, 14, 17, 30, 40, and 67) associated with *Tg-ABCs* displayed limited collinear gene pairs with genes on other scaffolds. In contrast, scaffolds without *Tg-ABCs*, specifically scaffold31 and scaffold36, exhibited higher numbers of collinear gene pairs with scaffold11 (Figure 3b). The integration of scaffold localization information and collinearity analysis results suggests that the functional roles of *Tg-ABCs* in *T. gibbosa* are diverse, potentially contributing differently to the process of wood degradation.

### 3.4. Physicochemical Analyses of Tg-ABCs in T. gibbosa

Physicochemical analyses (Table 1) revealed that the 12 Tg-ABC proteins in *T. gibbosa* share comparable lengths and molecular weights. The average amino acid length is 1521.5, and the average molecular weight is 167,619.4 Da, ranging from 145,732.30 to 185,461.04 Da. These proteins exhibit closely comparable lengths and molecular weights, displaying fluctuations around their respective averages.

Theoretical isoelectric points for Tg-ABC proteins range from 5.85 to 8.51, and the aliphatic index varies between 86.22 and 107.98. The average protein hydropathicity spans from −0.065 to 0.262, with the mean hydropathicity of five proteins being negative, classifying them as hydrophilic. In contrast, the remaining seven proteins exhibit positive predicted values, indicating their hydrophobic nature. These proteins are composed of five atoms, namely C, N, H, O, and S. Notably, three proteins (gene_5291, gene_11540, and gene_9723) have an instability index exceeding 40, suggesting structural instability, while the majority of proteins have an index below 40, indicating stable structures.

### 3.5. Phylogenetic Relationship and Collinearity Analyses of ABC Gene Family

To elucidate the similarity between *Tg-ABCs* and homologous genes in other species, we selected four wood-rotting fungi (*T. versicolor*, *G. leucocontextum*, *L. sulphureus*, and *P. centrifuga*) with comparable subfamily classifications of ABC genes to *T. gibbosa*. Subsequently, we screened the protein sequences of ABC subfamilies (ABCA, ABCB, ABCC, and ABCG) from each of these fungi (Appendix A) and constructed a collinearity relationship between *T. gibbosa* and each of the other species (Figure 4).

We conducted a phylogenetic analysis of *T. gibbosa* and four wood-rotting fungi (Figure 4a). We observed that in the phylogenetic trees of each ABC subfamily, the ABC genes of *T. gibbosa* and those of its same-genus fungus (*T. versicolor*) are positioned on relatively close branches. Additionally, the branches of ABC genes within the same subfamily are closer than those within the same species, indicating a higher degree of homology within the same ABC subfamily across different species. This suggests that ABC protein sequences exhibit a higher level of conservation during the evolutionary process.

We individually constructed collinearity relationships for *T. gibbosa* and four wood-rotting fungi (Figure 4b). In the collinearity analysis between *T. gibbosa* and its same-genus fungus (*T. versicolor*), we observed that among the twelve *Tg-ABCs*, nine exhibited collinearity, with seven, four, and two *Tg-ABCs* displaying collinearity with the other three species, respectively. Integrating the results from the phylogenetic and collinearity analyses, it becomes evident that *T. gibbosa* shares a closer phylogenetic relationship with *T. versicolor*, followed by *G. leucocontextum*, *L. sulphureus*, and *P. centrifuga* in descending order of phylogenetic proximity.

### 3.6. Quantitative Analysis of Tg-ABCs in T. gibbosa

To investigate the response of *T. gibbosa*’s *Tg-ABCs* to woody and non-woody environments, we conducted a comparative analysis, examining the relative expression levels of *Tg-ABCs* at different culture times in these distinct conditions (Figure 5). The results reveal that regardless of the cultivation duration, there are significant differences in the expression levels of *Tg-ABCs* in woody and non-woody environments. Specifically, at 10 days, the majority of genes exhibited noticeable downregulation, while the upward trends of other genes were also evident, indicating the initiation of an interaction between *T. gibbosa* and wood. This suggests the involvement of *Tg-ABCs* in the wood degradation process. Between days 15, 20, and 25, there were highly significant trends of both upregulation and downregulation for all 12 *Tg-ABCs*, with peak values observed during this period. This signifies that wood degradation is most strongly influenced by *Tg-ABCs* during this timeframe. At day 30, a relatively calm period ensues, with minimal changes in the expression levels of *Tg-ABCs*, suggesting a potentially minor impact of *Tg-ABCs* on wood degradation during this stage. However, at day 35, the previous calm period is disrupted, and new fluctuations in the expression levels of *Tg-ABCs* emerge, indicating a renewed and more significant impact of *Tg-ABCs* on wood degradation at this stage.

To further investigate the changes in the relative expression levels of *Tg-ABCs* in response to gradients of *T. gibbosa*’s wood degradation time, we analyzed the relative expression levels of *Tg-ABCs* cultivated in woody environments for 10, 15, 20, 25, 30, and 35 days (Figure 6). The results reveal that during a cultivation period of *T. gibbosa* in a woody environment lasting from 10 to 35 days, the relative expression levels of most *Tg-ABCs* followed a trend of increasing, decreasing, and increasing, with peak values observed at 20 and 35 days. At these time points, *Tg-ABCs* may have the most significant impact on *T. gibbosa*’s wood degradation. The relative expression levels of *gene11539* and *gene11540* showed a trend of increasing and then decreasing, with one fewer increase compared to most genes. *Gene8398* demonstrated a trend of increasing, decreasing, increasing, and then decreasing again, featuring one additional decrease compared to most genes. On the other hand, *gene5291* exhibited one additional increase compared to *gene8398*. As for *gene6080*, its expression had an extra decrease in the early stage compared to the majority of genes. These variations in expression trends may stem from differences in the functions of distinct genes, and different ABC subfamily genes may also play specific roles in wood degradation.

## 4. Discussion

The investigation of wood degradation holds substantial ecological and biological importance. Efficient wood degradation is vital for carbon cycling in ecosystems and provides essential technical support for sustainable biofuel production [35,36]. The ABC gene family plays a crucial role in lignin, cellulose, and hemicellulose metabolism, regulating vital substrate transport during this process [37]. *T. gibbosa*, as one of the numerous wood-rotting fungi, demonstrates efficient wood degradation capabilities [38], presenting a promising opportunity to investigate the role of the ABC gene family in wood degradation.

We identified a total of 12 *Tg-ABCs* by analyzing the genomic sequence of the ABC gene family in *T. gibbosa*. The results of the DNA binding domain alignment reveal that all of these genes contain conserved ABC transporter domains, exhibiting a high degree of sequence conservation. *Tg-ABCs* are classified into four subfamilies, ABCA, ABCB, ABCC, and ABCG. These subfamilies likely play significant roles in wood degradation, while the absence of five subfamilies [39] may have a relatively minor impact on this process. Scaffold localization results indicate that *Tg-ABCs* are dispersed on scaffolds. A collinearity analysis reveals that there is no duplication of gene sequences within the *Tg-ABCs* in the genome sequence of *T. gibbosa*. These findings collectively suggest that the functions of *T. gibbosa*’s ABC gene family in the wood degradation process may exhibit diversity. Phylogenetic and collinearity analyses of *T. gibbosa* with four other wood-rotting fungi show that *T. gibbosa* shares a closer phylogenetic relationship with its same-genus fungus (*T. versicolor*) [40], followed by *G. leucocontextum*, *L. sulphureus*, and *P. centrifuga* in descending order of phylogenetic proximity. Additionally, the branches of ABC genes within the same subfamily are closer than those within the same species, indicating a higher degree of homology within the same ABC subfamily across different species. This suggests that ABC protein sequences exhibit a higher level of conservation during the evolutionary process, and their wood degradation characteristics may also show a higher degree of similarity [41].

To gain a deeper understanding of how *Tg-ABCs* response to wood degradation by *T. gibbosa*, we conducted quantitative analyses of the expression of *Tg-ABCs* from *T. gibbosa* cultivated in both woody and non-woody environments for 10, 15, 20, 25, 30, and 35 days using an RT-qPCR analysis. The results reveal a significant difference in the expression levels of *Tg-ABCs* between woody and non-woody environments, suggesting an active involvement of the ABC gene family in wood degradation [21,42,43]. During the wood degradation period of *T. gibbosa*, spanning from 10 to 35 days, the relative expression levels of most *Tg-ABCs* exhibited a trend of increasing, decreasing, and then increasing again. It is speculated that this trend is associated with the different products generated during various stages of wood degradation by *T. gibbosa*. In the early stages of wood degradation, *T. gibbosa* produces a significant amount of extracellular enzymes, and ABC transporters facilitate the transport of these enzymes from the extracellular space into the cell, leading to an increase in their expression levels. In the mid-phase of wood degradation, these enzymes catalyze the oxidation and decomposition of large molecules such as lignin, cellulose, and hemicellulose. At this stage, ABC transporters are not required, resulting in a decrease in their expression levels. In the later stages of wood degradation, the generation of small molecular substances like acids, alcohols, and esters prompts ABC genes to transport these substances, causing a resurgence in their expression levels [16,44,45]. The fluctuation in the relative expression levels of *Tg-ABCs* corresponds to the metabolic trajectory of wood degradation, indicating the pivotal role of *Tg-ABCs* in the process. Additionally, at 20 and 35 days of wood degradation by *T. gibbosa*, the relative expression levels of *Tg-ABCs* peak, suggesting that at these time points, *Tg-ABCs* exert the most significant impact on wood degradation by *T. gibbosa*.

## 5. Conclusions

In this study, we systematically analyzed the ABC gene family in *T. gibbosa*, examining its structure, predicting its functions, and identifying the characteristics of *Tg-ABCs*. Additionally, we also explored how *Tg-ABCs* respond to woody environments. Our findings revealed that the ABC gene family plays a crucial role in the degradation of poplar wood by *T. gibbosa*. This study lays the foundation for a deeper understanding of the wood degradation mechanism of the ABC gene family and provides strong support for the development of more efficient strategies for the biological treatment of wood and other organic materials.

## Figures and Tables

**Figure 1 jof-10-00096-f001:**
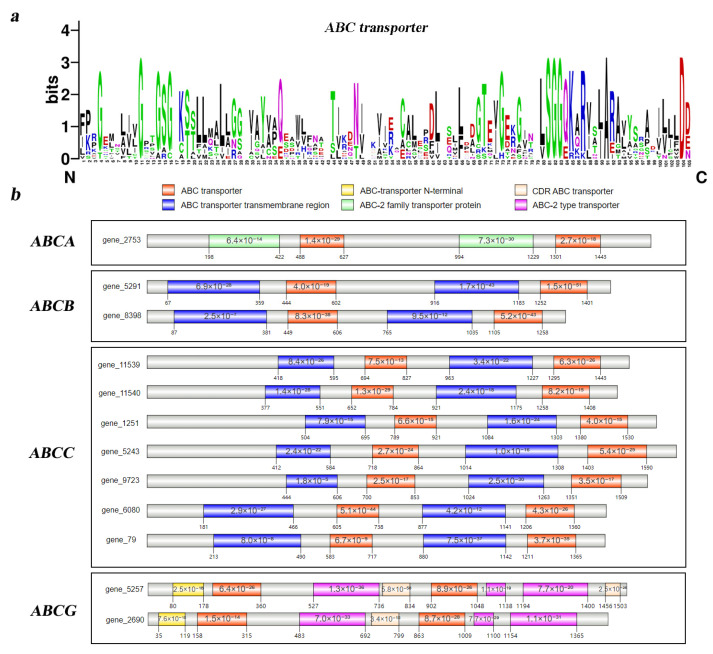
The DNA binding domain analysis of the ABC gene family in *Trametes gibbosa*. (**a**) DNA binding domain recognition marker. The symbols represent amino acids, and the height of each symbol within the stack indicates the relative frequency of that amino acid at the corresponding position; (**b**) DNA binding domain and subfamily classification of *Tg-ABCs*. Distinct domains are differentiated by different colors, with numerical values on each domain representing *p*-values. The numbers on the lower left and right sides of each domain represent the starting and ending positions of the respective domain.

**Figure 2 jof-10-00096-f002:**
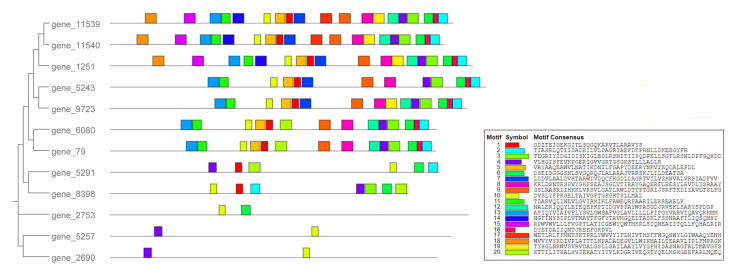
The phylogenetic tree and motifs of *Tg-ABCs* in *Trametes gibbosa*. Boxes of different colors represent different motifs.

**Figure 3 jof-10-00096-f003:**
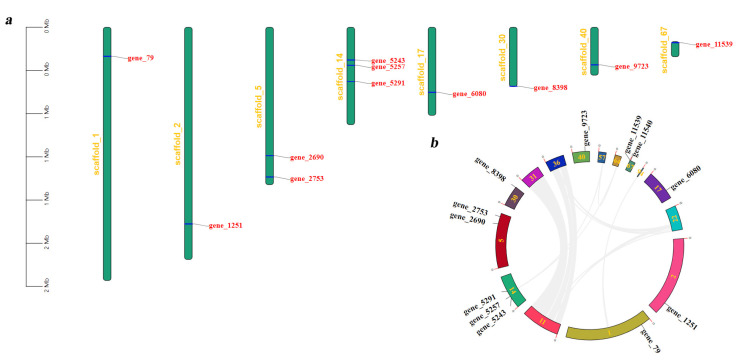
Scaffold localization and collinearity analyses of *Tg-ABCs* in *Trametes gibbosa*. (**a**) Distribution of *Tg-ABCs* on scaffolds; (**b**) segmental duplication events of *Tg-ABCs*. Displayed here are only the scaffolds that contain specific genes or collinear gene sequences.

**Figure 4 jof-10-00096-f004:**
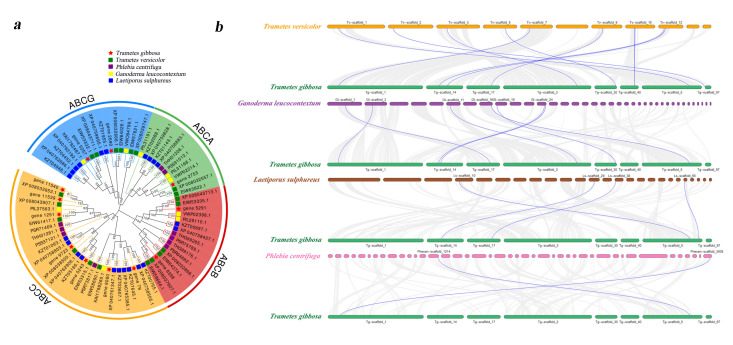
Phylogenetic tree and collinearity analyses of the ABC gene family. (**a**) Phylogenetic analysis of ABC proteins in *Trametes gibbosa, Trametes versicolor, Ganoderma leucocontextum, Laetiporus sulphureus, and Phlebia centrifuga*. ABC genes in *T. gibbosa* are denoted with red pentagrams, while those in the other four wood-rotting fungi are indicated with differently colored squares. Numerical values on branches represent the confidence level of each branch, with higher values indicating increased reliability; (**b**) collinearity relationships between genes of *T. gibbosa* and four wood-rotting fungi. The blue lines represent collinearity among ABC genes in *T. gibbosa* and other species, while the grey lines represent orthologous collinearity between the genome of *T. gibbosa* and the genomes of other species.

**Figure 5 jof-10-00096-f005:**
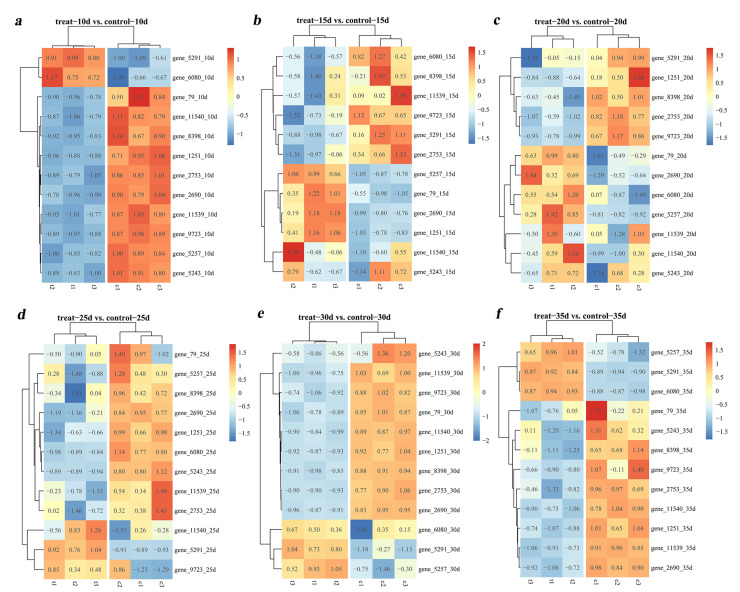
Heatmap of the relative expression levels of *Tg-ABCs* in *Trametes gibbosa* in woody and non-woody environments. The ‘treat-10d’ represents the expression levels of *Tg-ABCs* following 10 days of wood chip supplementation, while ‘control-10d’ indicates the expression level of *Tg-ABCs* without the addition of wood chips for 10 days. The series continues with ‘(**a**–**f**)’, representing the expression levels of *Tg-ABCs* at 10, 15, 20, 25, 30, and 35 days, respectively.

**Figure 6 jof-10-00096-f006:**
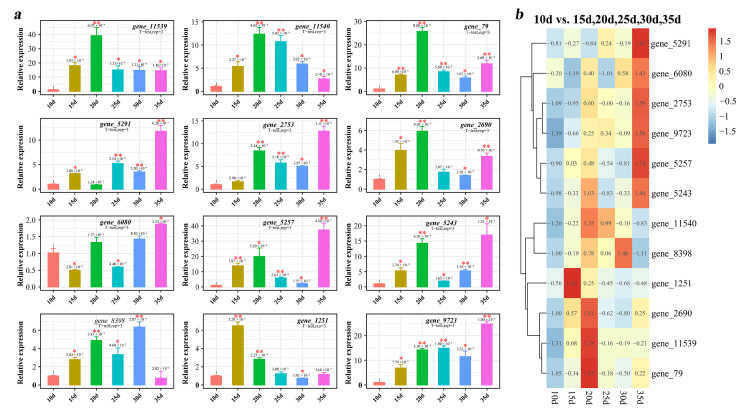
The relative expression analysis of *Tg-ABCs* in *Trametes gibbosa* under different cultivation time gradients. (**a**) Histogram of the relative expression analysis of *Tg-ABCs.* ** represents highly significant (*p* ≤ 0.01), * represents significant (0.01 < *p* < 0.05); (**b**) heatmap of the relative expression analysis of *Tg-ABCs*.

**Table 1 jof-10-00096-t001:** Physicochemical properties of Tg-ABC proteins in *Trametes gibbosa*.

ID	Number of Amino Acids	Molecular Weight	Theoretical pI	Aliphatic Index	Grand Average of Hydropathicity	Formula	Total Number of Atoms	Instability Index	Stability
gene_11539	1536	169,596.99	6.28	107.98	0.186	C_7664_H_12155_N_2041_O_2200_S_47_	24,107	37.98	stable
gene_11540	1497	165,312.94	5.92	106.73	0.117	C_7435_H_11789_N_1999_O_2176_S_41_	23,440	40.53	unstable
gene_79	1459	161,918.99	7.99	95.42	−0.030	C_7256_H_11479_N_1993_O_2107_S_49_	22,884	35.49	stable
gene_5291	1476	161,417.17	6.84	97.24	0.104	C_7183_H_11477_N_1997_O_2095_S_65_	22,817	48.30	unstable
gene_2753	1604	174,825.02	8.45	105.57	0.262	C_7940_H_12497_N_2113_O_2241_S_44_	24,835	35.36	stable
gene_2690	1465	164,291.53	6.96	86.22	−0.065	C_7446_H_11457_N_1969_O_2109_S_62_	23,043	39.54	stable
gene_6080	1463	160,933.00	8.51	97.58	−0.019	C_7214_H_11504_N_1958_O_2122_S_41_	22,839	36.87	stable
gene_5257	1524	168,164.73	7.00	86.71	−0.018	C_7560_H_11723_N_2059_O_2194_S_50_	23,586	37.30	stable
gene_5243	1685	185,461.04	5.89	101.38	0.082	C_8390_H_13216_N_2228_O_2432_S_39_	26,305	36.75	stable
gene_1251	1622	178,758.07	5.85	104.06	0.156	C_8094_H_12776_N_2108_O_2350_S_48_	25,376	39.28	stable
gene_9723	1594	175,020.65	5.90	102.61	0.169	C_7909_H_12477_N_2083_O_2292_S_50_	24,811	41.41	unstable
gene_8398	1333	145,732.30	6.24	96.05	−0.051	C_6518_H_10338_N_1774_O_1951_S_30_	20,611	39.62	stable

## Data Availability

The original contributions presented in the study are included in the article/Appendix A, further inquiries can be directed to the first author.

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
