# Peer review of "Genome-Wide Identification of the ABC Gene Family and Its Expression in Response to the Wood Degradation of Poplar in Trametes gibbosa"

_jof, 2024, doi:10.3390/jof10020096_

Round 1
Reviewer 1 Report
Comments and Suggestions for Authors
The paper is basically OK. I did not find scientific mistakes, but I added several remarks/questions in the corrected file.

Author Response
Dear Reviewer,
Thanks very much for taking your time to review our manuscript entitled "Genome-wide identification of the ABC gene family and its expression in response to wood degradation of Poplar in Trametes gibbosa" (Manuscript ID: jof-2814780). Your comments have proven to be invaluable and highly beneficial for the revision and improvement of our manuscript. We have carefully reviewed all your comments and made the necessary revisions to the manuscript, hoping they align with your expectations. The revised manuscript has been uploaded, with changes made using Word's track changes and marked in red.
Additionally, we have organized the inquiries you raised, summarizing them into several main questions. The attached PDF contains our detailed point-by-point response to these questions.
Best regards!

Reviewer 2 Report
Comments and Suggestions for Authors
Specific comments:
Line 101 – Please add strain number.
Line 110– Please add citation for medium.
Line 112 – Please add size of the wood chips.
Line 115– How the wood chips were removed? Please explain.
Line 117 – The paper is lacking information on how DNA was isolated, how genome/genes were sequenced.
Line 135 – Which genome? Accession number?
Line 154 – The paper is lacking information on how RNA was isolated, RT-PCR set-up, kits etc….
Line 160 – Why 10-day culture was chosen as control experiment?
Line 163 – Which genome?
Line 278 – Why the genes were downregulated?
Author Response
Dear Reviewer,
Thanks very much for taking your time to review our manuscript entitled "Genome-wide identification of the ABC gene family and its expression in response to wood degradation of Poplar in Trametes gibbosa" (Manuscript ID: jof-2814780). Your comments have proven to be invaluable and highly beneficial for the revision and improvement of our manuscript. We have carefully reviewed all your comments and made the necessary revisions to the manuscript, hoping they align with your expectations. The revised manuscript has been uploaded, with changes made using Word's track changes and marked in red.
Additionally, the attached PDF contains our detailed point-by-point response to your comments.
Best regards!
